# Socio-Demographic Factors Influencing Malaria Vaccine Acceptance for Under-Five Children in a Malaria-Endemic Region: A Community-Based Study in the Democratic Republic of Congo

**DOI:** 10.3390/vaccines12040380

**Published:** 2024-04-02

**Authors:** Arsene Daniel Nyalundja, Patrick Musole Bugeme, Ashuza Shamamba Guillaume, Alain Balola Ntaboba, Victoire Urbain Hatu’m, Jacques Lukenze Tamuzi, Duduzile Ndwandwe, Chinwe Iwu-Jaja, Charles S. Wiysonge, Patrick D. M. C. Katoto

**Affiliations:** 1Center for Tropical Diseases and Global Health (CTDGH), Catholic University of Bukavu (UCB), Bukavu 285, Democratic Republic of the Congo; nyalu_arse.adn@outlook.com (A.D.N.); musolebugeme@gmail.com (P.M.B.); 2Faculty of Medicine, Catholic University of Bukavu (UCB), Bukavu 285, Democratic Republic of the Congo; shamambaguillaume@gmail.com (A.S.G.); balola.ntabobal1@gmail.com (A.B.N.); victoirehat@gmail.com (V.U.H.); 3Department of Epidemiology, Johns Hopkins Bloomberg School of Public Health, Johns Hopkins University, Baltimore, MD 21205, USA; 4Department of Global Health, Faculty of Medicine and Health Sciences, Stellenbosch University, Cape Town 7505, South Africa; drjacques.tamuzi@gmail.com; 5Cochrane South Africa, South African Medical Research Council, Cape Town 7501, South Africa; duduzile.ndwandwe@mrc.ac.za (D.N.); chinwelolo@gmail.com (C.I.-J.); sheyc@who.int (C.S.W.); 6Vaccine-Preventable Diseases Program, World Health Organization Regional Office for Africa, Brazzaville P.O. Box 06, Congo; 7Centre for General Medicine and Global Health, Department of Medicine, University of Cape Town, Cape Town 7505, South Africa

**Keywords:** plasmodium falciparum, prevention and control, BeDS, immunization, Bukavu

## Abstract

Two novel vaccines against malaria are proposed as a complementary control tool to prevent and reduce *Plasmodium falciparum* related disease and death in under-five children from moderate to high malaria transmission regions. The Democratic Republic of Congo (DRC) has committed to eradicate malaria by 2030, and significant efforts have been deployed to strengthen control and elimination measures. We aimed to understand factors influencing the malaria vaccine acceptability among the general population in eastern DRC. We conducted a survey among adult Congolese in Bukavu in March 2022. The questionnaire was adapted from the Behavioral and Social Drivers of vaccine uptake (BeSD) framework and was administered online and physically. Multivariate logistic regressions were built, and estimates were represented as adjusted odds ratios (aOR) and corresponding 95% confidence intervals (95%CI). Out of 1612 adults (median age: 39 years, 46.15% female) surveyed, only 7.26% were aware of the malaria vaccine. However, 46.53% expressed willingness to vaccinate themselves, and 52.60% were open to vaccinating their under-five children. Adjusting for confounding factors, non-student/non-healthcare worker professions (aOR = 0.58, 95%CI [0.42–0.78]) and middle-income status (aOR = 1.87, 95%CI [1.25–2.80]) were significantly associated with self-vaccination acceptance. Age played a role in under-five child vaccination acceptability, with 25 to over 64 years showing increased acceptability compared to the 18–24 age group. Additionally, non-student/non-healthcare worker professions (aOR = 1.88, 95%CI [1.37–2.59]), medium education levels (aOR = 2.64, 95%CI [1.29–5.79]), and residing in semi-rural areas (aOR = 1.63, 95%CI [1.27–2.10]) were predictors of under-five child vaccination acceptance. The acceptability of the malaria vaccine for self and for under-five children was suboptimal for effective malaria control in this community in the DRC. Our study constitutes a call for the Expanded Program on Immunization to closely work with various stakeholders to strengthen risk communication for community engagement prior to and during the introduction of this novel and lifesaving tool, malaria vaccination.

## 1. Introduction

The Democratic Republic of Congo (DRC) is a central African country with an estimated 96 million inhabitants [1], experiencing malaria as a major public health problem ranking as the leading cause of disability and mortality combined [2,3]. It is estimated that the DRC has the second highest prevalence and mortality rate of malaria. The country shares 12% of the global burden [4], with 96.7% of households being malarious [5], and a prevalence of 13,246 cases per 100,000 population at risk over the two last decades [6]. Consequently, malaria accounts for 44% of all outpatient visits [3]; 10.15% (95% confidence interval, 4.70–16.95%) of overall deaths; 26.10% (12.66–41.36%) of under-five mortality with 81.1 deaths per 1000 live births [7]; 12.83% (6.35–20.55%) and 21.19% (12.36–39.62%) of total disability-adjusted life-years (DALYs) at all ages and under-five years, respectively [8]. Plasmodium falciparum is responsible for the majority of severe cases and is the most common malaria-induced parasite in the DRC [3,4]. However, cases of *Plasmodium malariae* and *Plasmodium ovale* are commonly reported in co-infection with *Plasmodium falciparum* [9].

To align with the global public health recommendation, the DRC has committed to eradicate malaria as per the “Zero Malaria Starts with Me” campaign launched since 2 July 2018, under the national theme: “I am committed to zero cases of malaria in my household!” [10]. The Ministry of Public Health, Hygiene, and Prevention exerts significant efforts to lower disease transmission and accelerate malaria elimination in the country. These efforts are implemented by the National Malaria Control Program and informed by the DRC Malaria Control Strategic Plan, which has subscribed to introduce malaria vaccination in its routine immunization program. It is expected that the use of vaccination as a supplementary control and elimination strategy will significantly reduce the burden of disease related to malaria and its socioeconomic burden in the DRC, already burdened with a decade of civil war.

RTS,S/AS01 vaccine (Mosquirix™), a protein-based recombinant malaria vaccine assessed in routine immunization programs in malaria-endemic countries [11] is the first vaccine targeting *Plasmodium falciparum* to be recommended by the World Health Organization (WHO) [12,13]. In 2017, Ghana, Kenya, and Malawi were selected to pilot its introduction, as well as to evaluate its effectiveness as a complementary tool to control malaria in endemic countries [4]. When combined with other known tools, such as oral antimalarial drugs, long-standing insecticide treated bed nets, indoor spraying of mosquito nets, and larvicide, RTS,S/AS01 was safe and effective in reducing both malaria severity and death. It is also cost-effective and feasible when implemented in routine immunization program in moderate-to-high malaria transmission areas [4,13]. To date, more than six million doses have been administered to over two million under-five children as part of the pilot Malaria Vaccine Implementation Program in Ghana, Kenya, and Malawi. Furthermore, the WHO recently recommended the R21/Matrix-M malaria vaccine as an additional safe and effective vaccine, offering a second choice to consumers [14]. This recommendation is expected to boost vaccine supply, making it more accessible and, consequently, leading to a significantly greater public health impact.

However, data examining factors influencing malaria vaccine uptake in many malarias’ endemic countries are very scanty. Considering the recently documented high prevalence rates of vaccine hesitancy that has significantly impeded the introduction of the COVID-19 vaccine across the world and particularly in the DRC, a comprehensive understanding of the community’s willingness to take malaria vaccine or to recommend it for their own children is instrumental to inform the introduction of any novel vaccine in a malaria endemic area. This study therefore aimed to assess malaria vaccine acceptability rate and associated drivers in the general population of the DRC to guide decisions and practices while scaling up the malaria vaccine.

## 2. Materials and Methods

### 2.1. Study Design, Period, and Setting

A community-based cross-sectional study was conducted among the adult population of Bukavu from 1 to 31 March 2022. This study forms a component of the Vaccine Confidence Project undertaken in Bukavu, which aimed to evaluate the acceptability of the general population towards vaccines for COVID-19, cholera, and malaria. Bukavu, a post conflict setting is the capital city of the South Kivu province in the eastern DRC, and administratively, it has three municipalities: Ibanda (urban), Kadutu (peri-urban), and Bagira (peri-urban). The city is located southwest of the Lake Kivu, and west of Cyagungu, Rwanda, from which it is separated by the Rizizi River (Figure 1). It has an estimated urban population of 1,190,000, as of 2022 [15]. Al adults (aged 18 and above) in Bukavu formed the target population. In each municipality, respondents were selected based on the criterion of one respondent per household and a minimum polling step of 15 households was deliberately chosen to select the next household as the most recent census was available.

### 2.2. Data Collection Tool

We used the adapted WHO’ behavior and social drivers of vaccines (BeSD) for data collection [16] (Appendix A). It was structured into three sections: background information, vaccination-related information, and knowledge and acceptability. All the survey items were translated into the local language, Swahili and back translated into English for consistency. Data submitted using KoboCollect (version 2022.2.3) was checked for consistency and completeness to ensure quality control by a second researcher, the study supervisor.

### 2.3. Outcome Measures

The primary outcome was the acceptability of taking the malaria vaccine for oneself and for under-five children. The outcome was assessed using this question: “Would you accept the malaria vaccine if offered and made available?” and “Would you accept the malaria vaccine for your under-five children if offered and made available for her/him?” Responses were recorded using a 5-point Likert Scale with “1” indicating strongly disagree, “2” for disagree, “3” for neutral, “4” for agree, and “5” for strongly agree. Subsequently, a positive response combined “4” and “5” and was scored “1”, while a negative response combined “1” to “3” and was scored “0”.

### 2.4. Independent Measures

Independent variables mainly included socio-demographic data. Respondents’ age was recorded as continuous variable at the time of survey. It was further stratified into five categories, including 18–24, 25–39, 40–54, 55–64, and s ≤ 65 years. Gender was reported as binary variable (male and female). Religion was reported as categorical variable, including Catholics, Protestants, Muslims, and others. Furthermore, Christianity combined Catholics and Protestants, while non-Christianity included Muslims and other religions. Respondents were asked if their religion accepted malaria vaccines. Unknow, and outright refusal were combined into “refusal of vaccines by religion”. Profession assessed status of respondents at the time when the survey was conducted and this included student, healthcare work, or neither healthcare worker neither student professions. Location was the living place of respondents which included three municipalities of Bukavu (Ibanda, Bagira, and Kadutu), as well as other municipalities. Those who resided outside Bukavu were asked to provide detailed information about their living location, including province, city. Urban location included Ibanda municipality, semi-rural location, as well as Kadutu and Bagire, and rural areas around Bukavu, South Kivu province. Level of education was categorized as low (including those who did not attend any teaching school and those who attended primary school); medium (education level included complete secondary school degree); and high educational levels (graduated with post-graduate degree, master, or doctorate). Lastly, monthly income was stratified as the average of monthly income earned by the respondent during the 12 last months, and respondents were asked to select the most appropriate category.

### 2.5. Sample Size

The minimal sample size was determined utilizing the single population proportion formula, assuming a 32.26% of malaria vaccine willingness for under-five children, a 95% confidence interval (CI) and a 5% margin of error as per the BeSD questionnaire requirement. The calculated sample size yielded 336. Assuming each municipality as a cluster and to correct for differences in design, the sample size was multiplied by a design effect of 3, resulting in a sample size of 1008. To account for potential non-response, a 10% margin was added, resulting in a final sample size of 1109.

### 2.6. Statistical Analysis

Descriptive statistics were used to summarize quantitative variables using median and interquartile range (IQR), and categorical variables using frequency and percentage. The results were presented as tables and figures. Logistic regression analysis was done to assess the relationship between independent variables and acceptability of malaria vaccine for self (parents/caregivers) and for under five children. All variables having a *p*-value ≤ 0.2 in the binary logistic regression were imported into a multivariable regression model to control for any possible confounding effect. Further variables known to be associated with the outcome as per epidemiological and clinical relevance (respondents’ location and religion) were also inserted in the model. Finally, predictors of acceptability were determined for an adjusted Odds Ratio (aOR) > 1 and *p* < 0.05. All *p*-values were two-sided. All statistical analyses were conducted in SPSS version 26 (IBM, Seattle, WA, USA) and R (version 4.2.2).

## 3. Results

### 3.1. Knowledge about Malaria Vaccine and Prevalence of Its Acceptability

Ninety-seven (5.68%) participants declined to provide consent for the survey, resulting in a response rate of 94.32%. A total of 1612 adults with a median age of 39 years (interquartile range: 26–54 years) were surveyed. Out of these, only 117 (7.26%, 95%CI 5.78–8.21%) respondents had heard about the malaria vaccine. Most of the people who had heard about malaria vaccine were between 18 and 24 years old, male, Christian and were neither students nor health workers. Almost half of the respondents, 750 (46.52%, 95%CI 44.10–49.01%) reported willingness to accept the malaria vaccine. The prevalence of malaria vaccine acceptability for self in the general population of Bukavu was highest among respondents aged 18–24 (50.36%), male (48.84%), those with a medium education level (59.57%), students (58.28%), whose religion accepts vaccines (53.21%), and those with a monthly income between USD 200–500 (58.06%). Most (929, 52.61%, 95%CI 46.41–58.81%) of the respondents were willing to vaccinate their children if the vaccine was available. Acceptability of malaria vaccine for children was highest among respondents aged 18–24 (60.86%), female (64.92%), those whose religion approves vaccination (58.90%), urban residents (58.87%), those with a high level of education (57.47%), healthcare workers (63.61%), and those with a monthly income between USD 200–500 (61.30%) (Table 1).

### 3.2. Factors Associated with Acceptability of Malaria Vaccine

Profession, monthly income, and religious acceptance of malaria vaccine were found to be associated with the acceptability of malaria vaccine for self, whereas age, gender, profession, and location were associated with the willingness to accept malaria vaccine for their under-five children. Respondents who had a monthly income of around USD 200–500 has an estimated aOR (95% IC) of1.87 (1.25–2.80) and whose religion accepts vaccines (aOR: 1.54 [1.20–1.99]) were more likely to be vaccinated compared to those having a monthly income less than USD 50 and those whose religion does not, respectively. Unfortunately, being neither students nor healthcare workers was less likely associated with the odds of accepting malaria vaccine for self (Table 2).

Conversely, factors influencing the willingness to vaccinate children under five years included being in the age groups of 25–39 (1.93 [1.63–3.26]), 40–54 (2.53 [1.58–4.48]), 55–64 (1.96 [1.16–3.38]), or over 64 years (2.07 [1.11–3.94]), relative to the reference group of 18–24 years. Additionally, those who were neither students nor healthcare workers (1.88 [1.37–2.59]) compared with healthcare workers, residents of semi-rural areas (1.63 [1.27–2.10]), and individuals with a medium level of education (2.64 [1.29–5.79]) compared to those with lower educational levels, were more likely to accept vaccination for this age group (Table 3).

## 4. Discussion

In this study, we assess the socio-demographic factors influencing the acceptability of the malaria vaccine among the general population of Bukavu, a city in the eastern DRC that has experienced post-conflict challenges. This assessment focused on both the willingness of individuals to accept the vaccine for themselves and for children under the age of five. Overall, respondents exhibited a modest level of acceptance regarding the malaria vaccine, with a significant proportion being unaware of its existence. Respondents whose religion accepts vaccines were more likely to accept vaccines for themselves and have children under-five vaccinated in the future. Furthermore, the study identified that older adults, non-students and non-healthcare workers, residents of semi-rural areas, and those with medium education levels were more inclined to vaccinate children under five.

In the context of high malaria endemicity, the relatively low to moderate acceptance of the malaria vaccine, coupled with a lack of awareness, raises significant concerns, especially considering the elevated fertility rate and the substantial impact of malaria on child morbidity and mortality. This suggests a need for increased public health education, community engagement, and awareness campaigns about the malaria vaccine to enhance its acceptance [17]. Additionally, the study highlights the impact of religious beliefs on vaccine acceptance. Respondents whose religion was supportive of vaccinations were more inclined to accept vaccines for themselves and were more likely to vaccinate their children under the age of five in the future [18]. This underscores the importance of engaging religious leaders and communities in vaccine advocacy, as their influence can significantly affect the attitudes and decisions of individuals regarding vaccination. In post-conflict settings in Africa, these challenges are further intensified. Conflict often disrupts healthcare systems and education, leading to gaps in vaccine awareness and accessibility. Engaging with local communities and religious leaders becomes even more crucial in these contexts, as they play a pivotal role in rebuilding trust in healthcare systems and influencing public perceptions towards vaccination. Their endorsement can significantly enhance vaccine acceptance and uptake, especially in areas where healthcare infrastructure is rebuilding [19].

The findings of this study also highlight critical demographic and occupational influences on the acceptability of vaccinating children under the age of five against malaria. A notable trend is the variation in vaccine acceptability across different age groups. Older adults, particularly those aged 25 and above, demonstrated a higher willingness to vaccinate children under five compared to the younger age group of 18–24 years. This variation could be attributed to a range of factors, including differing levels of exposure to healthcare information, varying degrees of perceived vulnerability to disease, or differences in parental and caregiving responsibilities across these age groups [20]. The role of occupation is also significant, with non-students and non-healthcare workers showing greater vaccine receptiveness, potentially reflecting varying health literacy levels. Importantly, the study indicates that individuals with a medium education level are more inclined to vaccinate children under five than those with lower education levels, suggesting that targeted educational interventions could significantly improve vaccine awareness and acceptance [21]. Furthermore, residing in semi-rural areas surprisingly correlates with higher vaccine acceptance, challenging common notions about rural vaccine hesitancy and highlighting the importance of tailored public health campaigns that address these complex factors to boost vaccine uptake among young children. However, this could also indicate a nuanced understanding and acceptance of vaccinations in semi-rural communities, possibly driven by their unique experiences with healthcare access and disease exposure.

To effectively address the challenges associated with the introduction of the malaria vaccine in malaria-endemic countries across Africa, a comprehensive and tailored approach is essential. This approach should emphasize the development of health education and communication strategies that cater to the diverse needs of different populations within these countries. Studies from various African nations, such as Uganda, Ghana, Nigeria, and others, have highlighted significant disparities in awareness and acceptability of the malaria vaccine [22,23,24]. Tailoring health education and communication strategies to target medically and informationally at-risk groups, using local languages and adapting content to various literacy levels, is crucial. These strategies should be designed to avoid complex medical jargon and potential misinformation that could foster distrust in health services or government initiatives. The focus should be on increasing knowledge about the vaccine’s schedule, availability, safety, and efficacy, thereby reducing misconceptions and enhancing acceptability, especially among caregivers of children under five [25].

The variation in vaccine acceptability and awareness rates across different African contexts necessitates a deeper understanding of the socio-demographic determinants of vaccine hesitancy. Factors such as income levels and religious beliefs have been identified as significant influencers in vaccine hesitancy decisions [25,26]. Therefore, it is recommended to implement comprehensive public health strategies that integrate vaccination efforts with other health interventions. These strategies should address concerns about vaccine safety, affordability, accessibility, efficacy misconceptions, and administration methods. Collaborative efforts between national immunization programs and other health bodies within these countries are pivotal. Such collaborations can leverage existing relationships to improve malaria control and mitigate its burden. The National Strategic Plans for Malaria Control in these countries should include goals to ensure vaccine availability, aim to vaccinate a substantial percentage of the target population, and educate a significant portion of the population about the vaccine’s benefits and safety. This multi-faceted approach is aligned with the broader objective of reducing the malaria burden and progressing towards the elimination of malaria in endemic African countries [27,28,29]. As the first doses of the malaria vaccine are expected to arrive in most of African malaria endemic countries during the last quarter of 2023, with countries starting to roll them out by early 2024, vaccine acceptability and awareness rates are substantial in this stage [30].

This study assessing factors associated with vaccine hesitancy against the malaria vaccine in the DRC presents several strengths and limitations. Its robust community-based cross-sectional design, assessing multiple diseases, enhances its relevance to public health decision-making. The meticulous adaptation of the WHO’s BeSD questionnaire and linguistic accuracy of survey items strengthen the validity of data collected. However, limitations include reliance on self-report of vaccine acceptability, potential oversimplification of attitudes with a Likert scale, non-response margin, and the inability to establish causation in this observational study. Hence, these findings offer valuable insights while necessitating a nuanced interpretation. Additionally, future studies shall differentiate several other factors not addressed in this study such as impact of parenthood and number of children, assess the impact of specific type of malaria vaccine and raison for accepting or heisting malaria vaccine.

## 5. Conclusions

In this study evaluating socio-demographic drivers of malaria vaccine acceptability in Bukavu, a post-conflict city in the eastern DRC, for both individuals and their children under five, we found a modest overall acceptance of the vaccine, with a significant portion of respondents unaware of its existence. Religious acceptance emerged as a positive influencer of vaccine acceptability, highlighting the importance of engaging religious leaders and communities in vaccination advocacy. Additionally, demographic factors played a significant role, with older adults, medium education levels, and residence in semi-rural areas demonstrating higher vaccine receptiveness. These findings emphasize the need for targeted public health education and awareness campaigns, particularly in post-conflict settings, to bridge gaps in vaccine knowledge and promote acceptance, ultimately contributing to malaria control efforts in the region.

## Figures and Tables

**Figure 1 vaccines-12-00380-f001:**
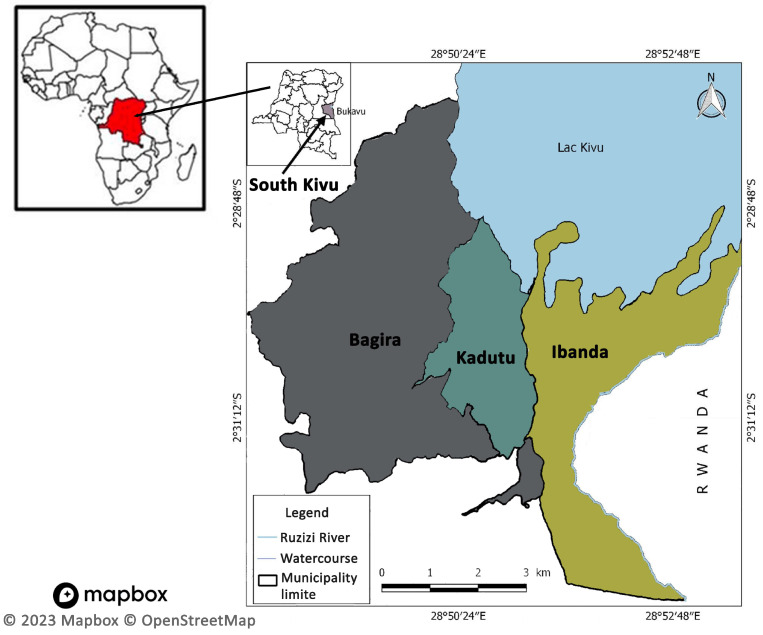
Administrative map of Bukavu including the three municipalities (Bagira, Kadutu, and Ibanda), and the bordering country (Rwanda). OpenStreetMap (CC BY-SA 2.0).

**Table 1 vaccines-12-00380-t001:** Distribution of malaria vaccine acceptability by respondent’s socio-demographic characteristics.

Characteristics		Total	Acceptability of Malaria Vaccine for Self (N, %)			Malaria Vaccine for Under-Five Children (N, %)		
			Acceptance	Hesitancy	*p*-Value	Acceptance	Hesitancy	*p*-Value
**Overall**		1612 (100%)	750 (46.53%)	862 (53.47%)		848 (52.61%)	764 (47.39%)	
**Age, years**					**0.01**			**<0.001**
	18–24	276 (17.12%)	139 (50.36%)	137 (49.63%)		168 (60.86%)	108 (39.14%)	
	25–39	536 (33.26%)	268 (50.00%)	268 (50.00%)		310 (57.84%)	226 (42.16%)	
	40–54	408 (25.31%)	174 (42.65%)	234 (57.35%)		187 (45.83%)	221 (54,17%)	
	55–64	278 (17.24%)	130 (46.77%)	148 (53.23%)		138 (49.64%)	140 (50.36%)	
	≥65	114 (7.07%)	39 (34.21%)	75 (65.78%)		45 (39.47%)	69 (60.53%)	
**Gender**					**0.04**			**0.01**
	Female	744 (46.15%)	326 (43.81%)	418 (56.19%)		483 (64.92%)	385 (35.08%)	
	Male	868 (53.85%)	424 (48.84%)	444 (51.16%)		365 (42.05%)	379 (57.95%)	
**Religion**					0.35			0.20
	Non-Christian	46 (2.85%)	25 (54.34%)	21 (45.66%)		29 (63.04%)	17 (36.96%)	
	Christian	1566 (97.15%)	725 (48.21%)	841 (51.79%)		819 (52.30%)	747 (47.70%)	
**Relig. Accept. Vac. ***					**<0.001**			**<0.001**
	Yes	859 (53.29%)	457 (53.21%)	402 (46.79%)		506 (58.90%)	353 (41.10%)	
	No	753 (46.71%)	293 (40.02%)	460 (59.98%)		342 45.42%)	411 (54.58%)	
**Location**					0.17			**<0.001**
	Semi-rural	880 (54.59%)	396 (45.00%)	484 (55.00%)		417 (47.39%)	463 (52.61%)	
	Urban	732 (45.41%)	354 (47.13%)	378 (52.87%)		431 (58.87%)	301 (41.13%)	
**Level of education**					**<0.001**			**<0.001**
	Low	44 (2.73%)	12 (27.27%)	32 (72.73%)		11 (25.00%)	33 (75.00%)	
	Medium	564 (34.98%)	336 (59.57%)	228 (40.43%)		260 (46.10%)	304 (53.90%)	
	High	1004 (62.29%)	510 (50.80%)	494 (49.20%)		577 (57.47%)	427 (42.53%)	
**Profession**					**<0.001**			**<0.001**
	No student/no healthcare worker	961 (59.61%)	380 (39.55%)	581 (60.45%)		442 (46.00%)	519 (64.00%)	
	Student	302 (18.74%)	176 (58.28%)	126 (41.72%)		184 (60.92%)	118 (39.08%)	
	Healthcare worker	349 (21.65%)	194 (55.59%)	155 (44.41%)		222 (63.61%)	127 (36.39%)	
**Monthly income, $ USD**					**<0.001**			**<0.001**
	<50	459 (40.15%)	158 (34.43%)	301 (65.57%)		202 (44.01%)	257 (55.99%)	
	50–200	457 (39.98%)	215 (47.05%)	242 (52.95%)		252 (55.15%)	205 (44.85%)	
	200–500	186 (16.27%)	108 (58.06%)	78 (41.94%)		114 (61.30%)	72 (38.70%)	
	>500	41 (3.60%)	20 (48.79%)	21 (51.21%)		22 (53.66%)	19 (46.34%)	

* Religious acceptance of vaccine. Bold: *p*-values < 0.05.

**Table 2 vaccines-12-00380-t002:** Determinants of Malaria Vaccine Acceptance Among Parents/Caregivers (for self) in Bukavu, South-Kivu, Democratic Republic of Congo.

Characteristics		aOR	95% CI		*p*
			Lower	Upper	
**Age, years**	18–24	**Reference**			
	25–39	0.83	0.51	1.35	0.44
	40–54	0.68	0.41	1.12	0.13
	55–64	0.96	0.58	1.61	0.88
	≥65	0.81	0.43	1.50	0.50
**Gender**	Male	**Reference**			
	Female	1.01	0.79	1.30	0.93
**Religion**	Christian	**Reference**			
	Non-Christian	1.67	0.73	3.88	0.22
**Location**	Urban	**Reference**			
	Semi-rural	1.04	0.81	1.33	0.79
**Religious acceptance of vaccine**	No	**Reference**			
	Yes	1.54	1.20	1.99	**<0.001**
**Level of education**	High	**Reference**			
	Low	0.61	0.28	1.23	0.17
	Medium	0.84	0.62	1.13	0.24
**Profession**	HCW	**Reference**			
	No Student No HCW	0.58	0.42	0.78	**<0.001**
	Student	-			
**Monthly Income, USD**	<50	**Reference**			
	50–200	1.23	0.91	1.67	0.18
	200–500	1.87	1.25	2.80	**<0.01**
	>500	1.21	0.61	2.40	0.58

Bold: *p*-values < 0.05.

**Table 3 vaccines-12-00380-t003:** Determinants of Willingness to Vaccinate Under-Five Children Against Malaria Among Parents/Caregivers in Bukavu, South-Kivu, Democratic Republic of Congo.

Characteristics		aOR	95% CI		*p*
			Lower	Upper	
**Age,** **years**	18–24	**Reference**			
	25–39	1.93	1.63	3.26	**0.01**
	40–54	2.53	1.58	4.48	**<0.001**
	55–64	1.96	1.16	3.38	**0.01**
	>= 65	2.07	1.11	3.94	**0.02**
**Gender**	Male	**Reference**			
	Female	1.18	0.92	1.52	**0.18**
**Religious acceptance of vaccination**	No	**Reference**			
	Yes	1.44	1.12	1.86	**<0.01**
**Profession**	Healthcare worker	**Reference**			
	No Student No Healthcare Worker	1.88	1.37	2.59	**<0.001**
	Student	-			
**Location**	Urban	**Reference**			
	Semi-rural	1.63	1.27	2.10	**<0.001**
**Level of education**	Low	**Reference**			
	Medium	2.64	1.29	5.79	**0.01**
	High	1.32	0.98	1.78	0.06
**Monthly Income, USD**	<50	**Reference**			
	50–200	0.94	0.69	1.27	0.46
	200–500	0.84	0.55	1.26	0.69
	>500	1.30	0.64	2.60	0.39

Bold: *p*-values < 0.05.

## Data Availability

The data and analytical code on which this article is based are available on request from the corresponding author (KDM) or the first author (ADN). To approve a request, it must be justified from a methodological point of view and receive the consent of all authors. All requests can be made after publication of this manuscript with no end date.

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
