# Peer review of "Socio-Demographic Factors Influencing Malaria Vaccine Acceptance for Under-Five Children in a Malaria-Endemic Region: A Community-Based Study in the Democratic Republic of Congo"

_vaccines, 2024, doi:10.3390/vaccines12040380_

Round 1

Reviewer 1 Report

Comments and Suggestions for Authors

This manuscript deals with acceptance of malaria vaccines in a urban area of CDR, an African country. The study is based on ~2000 forms, but only ~6% are informed on malaria vaccines. The inquiries are based on the respondent population but 94% of the population are not aware of any malaria vaccine, thus they are answering on basis of a potential existing malaria vaccine,  thus the result is biased by their knowledege on the importance of malaria vaccine. This is easily accomplished by the fact that the groups of this society that are interested in the malaria vaccine is the more affected by the disease or more exposed to disease, as medium income medium educated people. The table 1 is constructed in a way that the information in obscured by the way of the percentages were calculated. Thus, the percent of the whole group in not important for the reader, but the participation inside the groups, for example in line level of education in the table, the data for vaccine acceptance/ofr hesitancy are presented as percent of the whole group as  

Low 44 (2.73%) 12 (0.74%) 32 (1.99%) 

Medium 564 (34.98%) 336 (20.84%) 228 (14.14%) )

High 1004 (62.29%) 510 (31.64%) 494 (30.65%) 

but in fact  the data must by presented as

low group  2,73% of respondents , 27% acceptancy and 73% hesitancy 

Medium 35% respondents with 60% acceptancy and 40% hesitancy 

High 63% of espondents, with 51% acceptancy and 49% of hesitancy. 

This table must be reconstructed and also the stataistics must be correctly presented.

The article was some complicated bias based on religions effect , performed in a institution with some religious links and those biases views could complicated the analysis. 

The article is important regarding the future acceptancy of a malaria vaccine in a complicated area, but the bias and informations must be corrected before adequate revision.

Thus, the article could be evaluated if tables were reconstructed and   and explaining why the look for religion in this group, before it could be considered for publication. 

Author Response

This manuscript deals with acceptance of malaria vaccines in a urban area of CDR, an African country. The study is based on ~2000 forms, but only ~6% are informed on malaria vaccines. The inquiries are based on the respondent population but 94% of the population are not aware of any malaria vaccine, thus they are answering on basis of a potential existing malaria vaccine,  thus the result is biased by their knowledge on the importance of malaria vaccine. This is easily accomplished by the fact that the groups of this society that are interested in the malaria vaccine is the more affected by the disease or more exposed to disease, as medium income medium educated people.

The table 1 is constructed in a way that the information in obscured by the way of the percentages were calculated. Thus, the percent of the whole group in not important for the reader, but the participation inside the groups, for example in line level of education in the table, the data for vaccine acceptance/ofr hesitancy are presented as percent of the whole group as  

Low 44 (2.73%) 12 (0.74%) 32 (1.99%) 

Medium 564 (34.98%) 336 (20.84%) 228 (14.14%) )

High 1004 (62.29%) 510 (31.64%) 494 (30.65%) 

but in fact  the data must by presented as

low group  2,73% of respondents , 27% acceptancy and 73% hesitancy 

Medium 35% respondents with 60% acceptancy and 40% hesitancy 

High 63% of espondents, with 51% acceptancy and 49% of hesitancy. 

This table must be reconstructed and also the stataistics must be correctly presented.

Reply

Thank you for your time to improve our work and for your valuable comments. Table 1 has been revised accordingly and appropriate changes have made (Line 207 – 215).

The article was some complicated bias based on religions effect, performed in a institution with some religious links and those biases views could complicated the analysis. 

The article is important regarding the future acceptancy of a malaria vaccine in a complicated area, but the bias and informations must be corrected before adequate revision.

Thus, the article could be evaluated if tables were reconstructed and   and explaining why the look for religion in this group, before it could be considered for publication. 

Reply

This study examines the influence of religion on vaccine uptake, focusing on contexts where religious leaders historically guide community decisions. Religion, a key social determinant of health, affects various aspects of life, including healthcare access and acceptability. Evidence suggests a strong correlation between vaccine uptake and religious beliefs, mediated by religious leaders. As demonstrated with covid-19 vaccine, we anticipated that in areas endemic with malaria, religion will significantly impact vaccination uptake through mechanisms such as decision-making, community engagement, and advocacy, particularly aiding vulnerable populations.

Reviewer 2 Report

Comments and Suggestions for Authors

The manuscript represents an exciting approach to health policy. However, several points in the manuscript require attention: 1) the questionnaire should be added in the supplementary file and translated into English. It requires validation, which was not mentioned in the text. 2) The use of vaccination in children, even if it is a question of the validated questionnaire, may require Ethical approval. As seen in the figure, the responses to vaccinating children are higher than self-vaccination, which generates a bias. Women were less prone to use self-vaccination? How many of the interviewed women had children? How many mothers hesitated to use the vaccine on their kids? Are their children's vaccination schemes up to date? How many males were fathers?

There are several mistakes in the tables that need to be checked. There are values like the non-Christian religion which require revision, a total of 56, but in the groups 25 and 21 then 29 and 17 ??? The statistical significance in the table is confusing. How many people who accept self-vaccination accept children's vaccination, and how many do not?  What are the main reasons for not vaccinating children? Why in the healthcare working group there is high hesitancy? What is the relationship between vaccine knowledge, general and specific, and hesitancy? Table 1 states that 17 % of the total people interviewed are in the first age span; however, in Tables 2 and 3, the data of this group is missing. Figure 2 is redundant. 

The discussion should be modified.

The text requires correction

Comments on the Quality of English Language

Moderate grammatical mistakes were found in the text.

Author Response

Comments and Suggestions for Authors:

The manuscript represents an exciting approach to health policy. However, several points in the manuscript require attention:

  • the questionnaire should be added in the supplementary file and translated into English. It requires validation, which was not mentioned in the text.

Reply

Thank you for your valuable comment. The questionnaire was adapted from the WHO BeDS COVID-19 vaccination tool, which the validation survey was conducted in 12 low-and-middle income countries, including the Democratic Republic of Congo, where our study was conducted. This questionnaire was published online and used in many countries during the COVID-19 pandemic to meet an urgent need (https://www.who.int/publications/i/item/WHO-2019-nCoV- ). The adapted question is provided as supplemental material  and changes has been made accordingly (line 132 and line 725).   

  • The use of vaccination in children, even if it is a question of the validated questionnaire, may require Ethical approval.

Reply

Thank you for your comment. We did not interview children but adults 18 and above. Further, this study received ethical approval from the Institutional review board of the Catholic University of Bukavu as highlighted in  lines 738 to 741.

  • As seen in the figure, the responses to vaccinating children are higher than self-vaccination, which generates a bias. Women were less prone to use self-vaccination?

Reply

Thank you for this important comment. The observation mentioned does not indicate bias; it reflects the situation in endemic regions where adults typically experience simple malaria, whereas children are prone to severe malaria, leading to a higher risk of mortality. This aligns with literature indicating vaccine acceptance correlates with the perceived risk of a disease, as noted in the complacency aspect of the 5C questionnaire.

  • How many of the interviewed women had children? How many mothers hesitated to use the vaccine on their kids? Are their children's vaccination schemes up to date? How many males were fathers?

Reply

Thank you for your insightful questions. In this study, we did not specifically collect data regarding the parental status of our respondents, including the number of children they have, parental hesitation towards vaccination, and the current status of their children's vaccination schemes. We acknowledge this as a limitation of our research and thus highlight an area for future research. Understanding the influence of parental status on vaccine acceptability could yield critical insights and assist in identifying further target groups for more effective interventions. We have noted this point for consideration in future studies and have addressed this aspect in our manuscript (lines 695 to 698), indicating our commitment to enhancing the depth of research in this area.

  • There are several mistakes in the tables that need to be checked. There are values like the non-Christian religion which require revision, a total of 56, but in the groups 25 and 21 then 29 and 17 ???

Reply

Thank you for your comment, we made appropriate change; the exact number was 46 not 56.

  • The statistical significance in the table is confusing. How many people who accept self-vaccination accept children's vaccination, and how many do not?  What are the main reasons for not vaccinating children?

Reply

In the context of our study area, where malaria significantly contributes to mortality among children under five, our findings revealed that 848 respondents (52.61%) supported vaccination for under-five children, while 764 (47.39%) were not in favor. These findings are detailed in Table 1. The scope of our research was specifically tailored to investigate the sociodemographic factors influencing the acceptance of the malaria vaccine. As a result, the study did not explore the underlying reasons behind the hesitance or refusal to vaccinate children against malaria. Recognizing this gap, we suggest that future studies should leverage this limitation to better understand drivers of malaria vaccine hesitancy and outright refusal in such endemic area.

  • Why in the healthcare working group there is high hesitancy?

Reply

Following the recommendation of Reviewer #1, we have revised Table 1 to present percentages by column vs row. A thorough examination of Table 1 now clearly shows that both self-vaccination hesitancy and hesitancy towards vaccinating children under five are significantly lower among healthcare workers. Thank you for this observation.

  • What is the relationship between vaccine knowledge, general and specific, and hesitancy?

Reply

Thank you for your comment. While this study addressed respondents' awareness of malaria vaccines, we did not evaluate their general or specific knowledge about malaria vaccines to this first study focused on socio-demographic drivers. This is an aspect that could be further elaborated on in future studies.

  • Table 1 states that 17 % of the total people interviewed are in the first age span; however, in Tables 2 and 3, the data of this group is missing.

Reply

Thank you for your comment. Frequency data for the sociodemographic characteristics in relation to acceptance/hesitancy were not reported in Tables 2 and 3 as they were already presented in Table 1 (see revised table 1).

  • Figure 2 is redundant. 

Reply

We have removed figure 2 and thank you for this external observation.

Reviewer 3 Report

Comments and Suggestions for Authors

Strengths

1.     The study’s aim to determine malaria vaccine acceptance is a very important public health issue to address.

2.     The authors provided a sufficient introduction (including great epidemiological data on the malaria burden in DRC, historical efforts to address malaria in DRC, etc.) to justify the need for this study.

3.     Great justification for the sample size calculation and adequate sample size.

Suggestions for improvement

 Major areas to improve are

1.     Just as the authors provided examples for the outcome variables, they should include predictors (independent variables) and their response options used in the study. Providing that information will be helpful to the readers to understand the study’s findings.

2.     The ethical clearance section is missing. Authors need to describe the IRB approval process and how they obtained informed consent from the study participants in the method section.

Minor areas.

1.     The authors need to provide citations for some factual statements. For example, these statements, “To date, more than 2.4 million doses have been administered to over 830 thousand children. Furthermore, the WHO recently recommended the R21/Matrix-M malaria vaccine as an additional safe and effective vaccine, offering a second choice to consumers.” need citations.

2.     The authors stated, “However, data examining factors influencing malaria vaccine uptake in many malarias’ endemic countries are very scary.” Is it scary or scanty?

3.     The authors stated, “… a comprehensive under-93 standing of the community's willingness to take malaria vaccine or to recommend it for their own children is instrumental to inform the introduction of any novel vaccine in COVID-19 endemic area.  Is it a COVID-19 endemic area or a malaria endemic area? This paper is about malaria so

Comments on the Quality of English Language

Can be improved

Author Response

Reviewer 3:

Comments and Suggestions for Authors

Strengths

  1. The study’s aim to determine malaria vaccine acceptance is a very important public health issue to address.
  2. The authors provided a sufficient introduction (including great epidemiological data on the malaria burden in DRC, historical efforts to address malaria in DRC, etc.) to justify the need for this study.
  3. Great justification for the sample size calculation and adequate sample size.

Overall comment: we are great full for commending our work.

Suggestions for improvement

Major areas to improve are

  1. Just as the authors provided examples for the outcome variables, they should include predictors (independent variables) and their response options used in the study. Providing that information will be helpful to the readers to understand the study’s findings.

Reply

Thank for your comment. Details on independents variables have been added. (Line 147 – 169).   

  1. The ethical clearance section is missing. Authors need to describe the IRB approval process and how they obtained informed consent from the study participants in the method section.

Reply

Thank you for your valuable comment. The ethical section is located on lines 683 to 868. The informed consent was obtained from participant before  the survey interview. Given the reluctance of most respondents to sign a printed informed consent form, we opted to procure verbal informed consent after briefing them on the study's purpose and objectives. The informed consent statement is outlined from lines 738 to 741.

Minor areas.

  1. The authors need to provide citations for some factual statements. For example, these statements, “To date, more than 2.4 million doses have been administered to over 830 thousand children. Furthermore, the WHO recently recommended the R21/Matrix-M malaria vaccine as an additional safe and effective vaccine, offering a second choice to consumers.” need citations.

Reply

Thank you for comment. The citation has been added (reference 14, line 91)

  1. The authors stated, “However, data examining factors influencing malaria vaccine uptake in many malarias’ endemic countries are very scary.” Is it scary or scanty?

Reply

Thank you. Appropriate change has been made as suggested (line 95).

  1. The authors stated, “… a comprehensive under-93 standing of the community's willingness to take malaria vaccine or to recommend it for their own children is instrumental to inform the introduction of any novel vaccine in COVID-19 endemic area. Is it a COVID-19 endemic area or a malaria endemic area? This paper is about malaria so

Reply

The current revised version has incorporated this change (line 99 to 100). Thank you.

Reviewer 4 Report

Comments and Suggestions for Authors

Thank you for sharing your article investigating socio-demographic factors influencing malaria vaccine acceptance among caregivers for children below the age of 5 years in a malaria-endemic region in the DRC. The following comments may help to improve the article.

L53-56: Try to avoid lengthy sentences for better readability; please revise.

L53-60:  May I suggest to re-write/sort this section for better readability, e.g., starting with global and country-wide facts and continuing with malaria facts among adults and children under 5 years of age.

L60-61: I believe "Plasmodium falciparum" should be italic. L79-80: Please check the correct spelling of P. falciparum throughout your manuscript.

L60-63: How is malaria confirmed in the DRC? 

L71: Can you be more specific which specific malaria vaccine will be implemented in the DRC's routine immunization program, i.e., stating RTS,S/AS01 already at this stage of your manuscript. 

L71-72: Please be more specific concerning the use of vaccines as a supplementary control and elimination strategy. What other measure(s) will be applied besides vaccination? 

L75-76: So will the vaccine be given in an at-risk or cohort manner? 

L82: For readers not in this field, please be more specific when stating "other known tools".

L86: Children of which age?

L94: In line with your manuscript, is "the community's willingness" or the parents' and caretakers' willingness? L97: Are you targeting the general population or parents and caretakers of children below 5 years of age? Please clarify and revise accordingly. 

L96: Is you focus on RTS,S/AS01 or R21/matrix-M or on both or unrelated? I am just wondering whether the decision to vaccinate a young child may be also product-related? 

L104: So the aim of your manuscript is not just on the malaria vaccine but also on cholera and COVID-19 vaccines or is the focus of the Vaccine Confidence Project?

L101-102 & 109: Is your target population really the adult population in your setting or the parent's and caretakers? Please clarify here and throughout your manuscript.

L111: How did you define a household?  

L120: Back transformed or translated? 

L124: Administering the malaria vaccine also to adults does not seem to be reflected well in your tile and the aim of your manuscript (L96-98).

L132-138: Is your sample size related to targeting children or children and their caretakers and parents? Also, please state the proportion you incorporated in your sample size calculation.

L146-148: Please state ethos variables associated with your outcome.

L151: Was no ethical clearance stated or is this section missing? 

L156: What kind of consent was obtained and from whom? How did you deal with potential illiteracy? 

Table 2 and 3: Please add the units for the variables age and monthly income. 

Figure 2B: How did you define urban and semi-urban? 

L286: Is your focus not just on malaria? Which diseases are your referring to?  

General comment: The discussion section could benefit from a more "lively" discussion of already published literature in this research field.  

Comments on the Quality of English Language

Moderate editing of English language required.

Author Response

Reviewer 4:

Comments and Suggestions for Authors

Thank you for sharing your article investigating socio-demographic factors influencing malaria vaccine acceptance among caregivers for children below the age of 5 years in a malaria-endemic region in the DRC. The following comments may help to improve the article.

L53-56: Try to avoid lengthy sentences for better readability; please revise.

Reply

Thank you for your comment. The sentence has been revised accordingly. (Line 55 to 58)

L53-60:  May I suggest to re-write/sort this section for better readability, e.g., starting with global and country-wide facts and continuing with malaria facts among adults and children under 5 years of age.

Reply.

This valuable comment has been considered. Thank you.  

L60-61: I believe "Plasmodium falciparum" should be italic. L79-80: Please check the correct spelling of P. falciparum throughout your manuscript.

Reply.

We have inserted this valuable comment. Thank you.  

L60-63: How is malaria confirmed in the DRC? 

Reply

In the DRC, malaria is confirmed following the national guidelines by RDT or microscopy (blood smear).

L71: Can you be more specific which specific malaria vaccine will be implemented in the DRC's routine immunization program, i.e., stating RTS,S/AS01 already at this stage of your manuscript. 

Reply.

Thank you. It will be the RTS,S/AS01. We have clarified accordingly. (line 73)

L71-72: Please be more specific concerning the use of vaccines as a supplementary control and elimination strategy. What other measure(s) will be applied besides vaccination? 

Reply

Thank you for your insightful comment. We have acknowledged this observation. Vaccine is part of the tools used in the country to combat malaria. Additional measures that have been implemented, with vaccines serving as a supplementary intervention, include the distribution of long-lasting insecticide-treated mosquito nets, initiatives to prevent malaria during pregnancy, enhancements in diagnostics and case management, intensified surveillance efforts, and the ongoing monitoring and evaluation of malaria-related activities.

L75-76: So will the vaccine be given in an at-risk or cohort manner?

Reply

Thank you for your inquiry.  Yes, according to WHO guidelines, countries in the WHO Africa region are categorized into three groups based on malaria endemicity, with the Democratic Republic of the Congo (DRC) classified as a high-endemicity country. Initially, the malaria vaccine rollout within these countries will prioritize at-risk groups, particularly children under five. Depending on the availability of vaccine doses beyond the Gavi Alliance framework, countries may extend eligibility to include older adults, visitors to endemic areas, and individuals with comorbidities or immunocompromised conditions.

L82: For readers not in this field, please be more specific when stating "other known tools".

Reply

Thank you for comment. Clarification has been made accordingly. (Line 84 to 86)

L86: Children of which age?

Reply

Thank you. To under-five children. Clarification has been made (line 89)

L94: In line with your manuscript, is "the community's willingness" or the parents' and caretakers' willingness?

Reply

Thank you for your valuable comment. The willingness of the community is paramount. Notably, as we included only adults in this multigenerational community, adults are often parents or caregivers.

L97: Are you targeting the general population or parents and caretakers of children below 5 years of age? Please clarify and revise accordingly. 

Reply

Thank you for your valuable comment. Our target population consisted of all adults (over 18 years) without regard to parenthood status.  (see response above)

L96: Is you focus on RTS,S/AS01 or R21/matrix-M or on both or unrelated? I am just wondering whether the decision to vaccinate a young child may be also product-related? 

Reply

Thank you for your comment. As we did with previous study addressing COVID-19 vaccine hesitance at the beginning of roll out. We did not focus on specific vaccines as they had not yet been integrated into the antimalarial intervention package in study setting.

L104: So the aim of your manuscript is not just on the malaria vaccine but also on cholera and COVID-19 vaccines or is the focus of the Vaccine Confidence Project?

Reply

Thank you for your valuable feedback. This manuscript aimed to assess the sociodemographic drivers of malaria vaccine acceptability. It stemmed from a larger project that evaluated the acceptability of both COVID-19 and non-COVID-19 vaccines in the post-COVID-19 era.

L101-102 & 109: Is your target population really the adult population in your setting or the parent's and caretakers? Please clarify here and throughout your manuscript.

Reply

Thank you for your valuable comment. Our target population comprised the general population of adults (over 18 years) without differentiation based on parenthood status. 

L111: How did you define a household?

Reply

We defined "household" as a group of people who live together in the same dwelling and share common spaces for cooking, eating, and living. A household may include two or more families, with related or unrelated individuals, with different housing but on the same parcel of land.

L120: Back transformed or translated? 

Reply

Thank you for the question. Is it back translated. Appropriate change has been made.

L124: Administering the malaria vaccine also to adults does not seem to be reflected well in your tile and the aim of your manuscript (L96-98).

Reply

Thank you for your valuable comment.  The title has been updated to indicate that adults are the target population in this community. Further, vaccinating adults is not the primary focus in this endemic malaria area, where adults typically experience only mild malaria cases, which seldom result in death, unlike in children.

L132-138: Is your sample size related to targeting children or children and their caretakers and parents? Also, please state the proportion you incorporated in your sample size calculation.

Reply

Our sample size was related to targeting caretakers and parents. The proportion included in sample size was 32.26%. Clarification has been made in main text (line 172).  

L146-148: Please state ethos variables associated with your outcome.

Reply

We apologize as we are not able to understand this question.

L151: Was no ethical clearance stated or is this section missing? 

Reply

Thank you for your valuable comment. The ethical clearance is mentioned on line 693 to 696.

L156: What kind of consent was obtained and from whom? How did you deal with potential illiteracy? 

Reply

We secured informed consent from every participant prior to their involvement in the study. To address potential challenges associated with illiteracy, we meticulously designed our survey questions using clear and straightforward language to facilitate comprehension. Additionally, we implemented verbal consent procedures where necessary, ensuring that all participants fully understood the study's purpose, procedures, and their rights before participation. Our data collectors (last year medical students) underwent extensive training to proficiently administer the survey, emphasizing the importance of clear communication and ensuring participants' understanding and comfort throughout the process. This approach ensured that all participants, regardless of their literacy level, could provide informed and voluntary consent.

Table 2 and 3: Please add the units for the variables age and monthly income. 

Reply

Thank you for comment. Appropriate changes have been made.

Figure 2B: How did you define urban and semi-urban? 

Reply

As recommended by other reviewers, we have deleted this figure.

L286: Is your focus not just on malaria? Which diseases are your referring to?  

Reply

yes, malaria. We have made this clear.

Round 2

Reviewer 1 Report

Comments and Suggestions for Authors

The authors corrected the percent use in table 1 which is the most important data from their Work. They also comment on the bias of the religion on the population opinion in the area, but they do not introduce formally the theme in the manuscript. They only provide data. It is a clever solution. 

I believe that additional comments on the subject would not provide any further improvement of the manuscript, thus I opt for accept the present form of the manuscript.  

Author Response

We genuinely appreciate your valuable feedback and are delighted to hear that you are pleased with the changes that have been made. Your inputs were highly valued throughout this process. Thank you for taking the time to share your thoughts with us.

Reviewer 2 Report

Comments and Suggestions for Authors

The authors have changed the text and incorporated the questionnaire as requested. The article is now suitable for publication

Comments on the Quality of English Language

minor grammatical mistakes were encountered.

Author Response

We are pleased to learn of your satisfaction with the changes made. Thank you. 

Reviewer 3 Report

Comments and Suggestions for Authors

Thanks for addressing the comments!

Author Response

We extend our sincerest gratitude for your invaluable feedback and are genuinely delighted to learn of your satisfaction with the recent changes. Your insights have been of immense importance throughout this endeavor, and we deeply appreciate the time and effort you dedicated to sharing your thoughts with us. Thank you wholeheartedly for your contribution.

We extend our sincerest gratitude for your invaluable feedback and are genuinely delighted to learn of your satisfaction with the recent changes. Your insights have been of immense importance throughout this endeavor, and we deeply appreciate the time and effort you dedicated to sharing your thoughts with us. Thank you wholeheartedly for your contribution.

Reviewer 4 Report

Comments and Suggestions for Authors

Thank you for sharing the revised manuscript. My comments were addressed sufficiently. 

Please also address my comment from the 1st review:

L146-148: Please state those variables associated with your outcome.

Author Response

L146-148: Please state those variables associated with your outcome

Reply

As per the literature, those variable were location and religion. We have also revised the manuscript to reflect this change. Thank you